# Epichaperome Inhibition by PU-H71-Mediated Targeting of HSP90 Sensitizes Glioblastoma Cells to Alkylator-Induced DNA Damage

**DOI:** 10.3390/cancers16233934

**Published:** 2024-11-24

**Authors:** Pratibha Sharma, Jihong Xu, Vinay K. Puduvalli

**Affiliations:** Department of Neuro-Oncology, The University of Texas MD Anderson Cancer Center, 1515 Holcombe Blvd., Houston, TX 77030, USA; psharma3@mdanderson.org (P.S.);

**Keywords:** heat shock protein 90 (HSP90), PU-H71, malignant glioma, HSP90 effector proteins, HSP70

## Abstract

Gliomas are malignant primary brain tumors recognized for acquiring resistance to conventional chemotherapy and radiotherapy. These tumors’ genetic heterogeneity and altered epigenetics contribute to their poor prognosis, underscoring the need to develop therapeutic strategies that target multiple pathways required for glioma cell survival and proliferation. Here, we tested the efficacy of a novel HSP90 inhibitor, PU-H71, in patient-derived glioma cells with various molecular subclassifications and in normal cells. PU-H71 strongly inhibited the proliferation, colony-forming ability, and migration of glioma cells. PU-H71-treated glioma cells showed a significant downregulation of HSP90 effector proteins that are essential for glioma cell survival and progression. PU-H71 induced significant programmed cell death in glioma cells but not normal cells, making it an ideal HSP90 inhibitor with low toxicity. These results establish that PU-H71 has potent activity against HSP90 in glioma cells and that further investigation of PU-H71 for the treatment of glioma is warranted.

## 1. Introduction

Malignant gliomas, the most prevalent type of brain tumors, are confirmed in 5–7 people per 100,000 annually [1,2]. Based on their histological and molecular features, the World Health Organization has classified gliomas into Oligodendroglioma IDH-mutant and 1p/19q-codeleted (WHO grade 2 or 3), Astrocytoma IDH-mutant WHO grade 2 or 3, Astrocytoma IDH-mutant Grade 4, Glioblastoma IDH wild type WHO grade 4, Diffuse hemispheric glioma, H3.3 G34 mutant, WHO grade 4 and Diffuse midline glioma, and H3 K27M-mutant WHO grade 4 [3]. Despite aggressive treatments, the prognosis of glioma remains poor owing to inter- and intratumoral heterogeneity [4], the limited blood–brain barrier (BBB) permeability of therapeutic agents, drug-resistant stem cell populations [5], and a lack of reliable biomarkers [4,6,7]. Targeting pro-survival pathways individually has failed to significantly improve the survival of glioma patients, indicating a need for therapeutic interventions that target several key survival pathways simultaneously [8,9]. Because they require a higher rate of building block synthesis than normal cells, glioma cells have a higher rate of nucleic acid and protein synthesis. Higher rates of protein synthesis require higher rates of protein processing. Therefore, targeting the key signaling molecules to halt the processing of proteins required for the survival of cancer cells is a viable glioma treatment strategy [10,11].

Heat shock proteins (HSPs) are a group of molecular chaperones that play a key role in the proper folding of newly synthesized polypeptide chains, membrane translocation of surface proteins, intracellular trafficking of proteins to various organelles, and proteasome-mediated degradation of misfolded proteins [11,12]. The genes that code for HSPs are evolutionarily conserved and are constitutively expressed in normal cells. However, cancer cells express HSPs at higher levels and tend to depend more on their client proteins for tumor maintenance and proliferation [10]. HSP90 is an ATP-dependent, highly conserved, ubiquitous molecular chaperone that regulates more than 200 client proteins involved in regulating the cell cycle (e.g., CDK4, CDK6), inducing angiogenesis (HIF1A, VEGFR, PI3K/AKT, RTKs), activating downstream pro-survival pathways (EGFR, PI3K/AKT, MAPK), deregulating cellular energies (ARNT, ARRB1, HIF1A, HMG1, SREBF1), and resisting cellular death (NF-kB, AKT, p53, c-MET, APAF1, survivin) [8,13,14,15]. The pharmacological inhibition of HSP90 simultaneously downregulates the expression of multiple effector proteins, making HSP90 an attractive therapeutic target in glioma [13,14,16,17]. Combining conventional therapies with HSP90 inhibitors has shown promising results in sensitizing glioma cells towards radiation [18], chemotherapy [17] or both [19].

HSP90 has two homologous isoforms, HSP90α and HSP90β, each of which has three major domains, a C-terminal domain for dimerization, a middle domain for client protein binding, and an N-terminal ATPase domain [20]. Several HSP90 inhibitors, such as 17-N-allylamino-17-demethoxygeldanamycin (17-AAG), 17-dimethylaminoethylamino-17-demethoxygeldanamycin (17-DMAG), geldanamycin, and radicicol, target the ATPase binding domain. However, in clinical trials, these drugs resulted in poor prognosis owing to toxicity and drug solubility issues, contributing to patients’ already poor outcomes [21,22,23,24,25]. An alternative to these drugs is the HSP90 inhibitor PU-H71, a novel, purine-based synthetic epichaperome inhibitor whose binding to ATP causes conformational changes that lock the client protein–HSP90 complex, inducing the proteasome-mediated degradation of client proteins [26]. Furthermore, owing to its higher binding affinity for HSP90-containing complexes, PU-H71 has a higher specificity for HSP90 in cancer cells than in normal cells, which reduces the overall toxicity of HSP90 inhibition [27]. The intravenous administration of PU-H71 in patients with refractory solid tumors over 1 h on days 1 and 8 of 21-day cycles was well tolerated (10 to 470 mg/m^2^/day) in this study, and no dose limiting toxicity was observed [28].

In this study, we investigated the efficacy of PU-H71 against the proliferation and other essential biological properties of patient-derived glioma stem-like cells (GSCs), adherent glioma cells, and normal human astrocytes (NHAs). We also studied the effect of PU-H71 treatment on the expression of EGFR and downstream pro-survival signaling effector molecules. We also evaluated the impact of PU-H71 in combination with the DNA-alkylating agent temozolomide in glioma cells.

## 2. Materials and Methods

### 2.1. Cell Lines and Reagents

The patient-derived GSC lines GSC262, GSC811, GSC11, GSC23, GSC20, and GSC272 were obtained from the MD Anderson Cancer Center cell repository. These cells were cultured as 3-dimensional neurospheres and subcultured in Dulbecco’s modified Eagle’s medium/F-12 nutrient mix (Life Technologies, Carlsbad, CA, USA, cat. #10565042) with 2% B-27 serum-free supplement (Life Technologies, cat. #17504-044), 0.02 µg/mL epidermal growth factor (Gold Biotechnology, Olivette, MO, USA, cat. #1150-04-100), 0.02 µg/mL basic fibroblast growth factor (Gold Biotechnology, cat. #1140-02-50), and 1% penicillin/streptomycin (Gibco, Baltimore, MD, USA, cat. #15140122) for 3–5 days. The human adherent glioma cell lines LN229 and T98G were purchased from the American Type Culture Collection and cultured in DMEM/F-12 supplemented with 1% penicillin/streptomycin (Gibco, cat. #15140122) and 5% fetal bovine serum at 37 °C. U251-HF glioma cells were kindly provided by Dr. W. K. Alfred Yung at MD Anderson and cultured as per the adherent glioma cell culture specifications. Human brain microvascular endothelial cells (HBMECs) were purchased from ScienCell Research Laboratories, Carlsbad, CA, USA (cat. #1000) and propagated according to the manufacturer’s instructions. NHAs were purchased from Lonza, Basel, Switzerland (cat. #CC-2565) and cultured according to the manufacturer’s instructions. For cell line authentication, DNA extracted from each cell line was sent to the University of Arizona Genetics Core at https://uagc.arl.arizona.edu/node/27 (accessed on 11 May 2024) for short tandem repeat analysis. PU-H71 was purchased from SelleckChem, Frankfurt am Main, Germany (cat. #S8039).

### 2.2. Cell Viability Assay

NHAs and GS262, GS811, GSC11, GSC23, GSC20, GSC272, LN229, T98G, and U251-HF cells were seeded in 96-well plates (2000 cells/well) and allowed to recover overnight. The cells were then treated with PU-H71 at concentrations ranging from 50 nM to 3.0 µM for 24–72 h. Cell viability was measured using a CellTiter-Glo kit (Promega, Madison, WI, USA, cat. #G7570) according to the manufacturer’s instructions. The relative light unit outputs for the PU-H71-treated cells were normalized to that for the vehicle-treated control cells and converted to percentages.

### 2.3. Cell Death Assay Using Annexin V and Propidium Iodide Staining

The GS811 and U251-HF neurospheres were treated with vehicle or PU-H71 (0.25 µM or 1.0 µM) for 48 or 72 h. The neurospheres were dissociated into a single-cell suspension. These cells were washed with phosphate-buffered saline, resuspended in 1× binding buffer, and then stained with annexin V–FITC and propidium iodide (BD Biosciences, Franklin Lakes, NJ, USA, cat. #556547). The treated cells were used for unstained, annexin-only, or propidium iodide-only controls to determine the population of viable, pre-apoptotic, apoptotic, and necrotic cells using a FACSCalibur flow cytometer.

### 2.4. Cell Cycle Analysis

The GSC811 and U251-HF cells were treated with vehicle or PU-H71 (0.25 µM or 1.0 µM). After treatment, the cells were harvested, and single-cell suspensions were prepared as described above. Single-cell suspensions of the treated cells were fixed and washed and then stained with propidium iodide using a kit (BD Biosciences, cat. #550825), according to the manufacturer’s instructions. The FL2-A histogram was used to identify the percentages of cells in the sub-G_1_, G_1_, S, and G_2_/M phase.

### 2.5. Colony Formation Assay

The LN229, T98G, and U251-HF cells were treated with vehicle or 1.0 µM PU-H71 for 24 h. Treated cells were washed with phosphate-buffered saline, and viable cells were plated in triplicate in 6-well plates (1000 cells/well). After 10–15 days, the media were removed, and the colonies were fixed with methanol and stained with crystal violet for 15–30 min. GelCount (Oxford Optronix, London, Canada) was used to count the stained colonies.

### 2.6. Wound-Healing Assay

The LN229, T98G, and U251-HF cells were seeded in triplicate in 12-well plates (10,000 viable cells/well). Once cell confluency reached 80%, unattached, floating cells were removed. Fresh media with a vehicle or 1.0 µM PU-H71 were added to the attached cells, and the cells were incubated for 24 h. Then, a straight-line scratch was created using a 200-µM pipette tip. The plate was imaged at 0 and 24 h using an AxioScope A1 (Zeiss, Jena, Germany). The area of no cell growth was quantified to assess the degree of wound-healing migration using ImageJ software 1.8.0 (https://imagej.nih.gov/ij/ accessed on 11 May 2024).

### 2.7. In Vitro Transwell Migration Assay

The impact of PU-H71 on endothelial cell migration was assessed using a protocol described previously [29]. Briefly, HBMECs were seeded in a 12-well plate (1.0 × 10^5^ cells/well) overnight. The attached cells were treated with vehicle or 2.0 µM PU-H71. Equal numbers of viable cells from the vehicle- and PU-H71-treated wells were added to inserts in a transwell plate (Corning, Corning, NY, USA, cat. #3460). The migrated cells were imaged using an AxioScope A1 microscope (Zeiss) and counted using ImageJ software 1.8.0 (https://imagej.nih.gov/ij/ accessed on 11 May 2024).

### 2.8. Immunoblotting

Immunoblotting was performed as described previously [19]. NHAs and the GSC262 and GSC811 cells were treated with vehicle or PU-H71 for 5–48 h. Cells were harvested, and whole-cell lysates were prepared using RIPA lysis buffer (Sigma, St. Louis, MO, USA, cat. #R0278) supplemented with protease and phosphatase inhibitor cocktails (Thermo Fisher Scientific, Waltham, MA, USA, cat. #78430). The total protein from each sample (30–40 µg) was subjected to electrophoresis on 4–20% gradient precast gel (Bio-Rad, Hercules, CA, USA, cat. #456-1094). The separated proteins were transferred to a nitrocellulose membrane using the Trans Blot Turbo Transfer System (Bio-Rad, cat. #1704150) and probed for HSP90, HSP70, cleaved PARP, GAPDH, p-AKT, AKT, pERK1/2, ERK1/2, pS6, S6, pEGFR, and EGFR. For the PU-H71 dose response, and for the PU-H71 time response, runs were performed by running the control and treated lysates on two separate gels. Blot-1 was sequentially probed for HSP70, AKT, p-AKT, pS6, and GAPDH (loading control). Parallelly, blot-2 was sequentially probed with cleaved-PARP, MAPK, p-MAPK, S6, and GAPDH (loading control). The activation of AKT and MAPK was compared using anti-rabbit p-AKT and p-MAPK and anti-mouse AKT and MAPK. S6 and pS6 were compared on separate membranes due to the unavailability of specific different species antibodies. Anti-mouse and anti-rabbit immunoglobulin G DyLight antibodies were used as secondary antibodies. Additional information about antibody amounts and manufacturers is provided in Appendix A.

### 2.9. Statistical Analysis

All experiments were conducted at least 3 times with at least 3 technical replicates in each experiment. The results of each experiment are presented as the mean ± SEM. Two-tailed *t*-tests and 1-way ANOVA were used to compare 2 sets of data and 3 or more sets of data, respectively. All graphs were created, and all statistical analyses were performed using GraphPad Prism 9 (GraphPad Software). In addition, *p* values less than 0.05 were considered to indicate significant differences.

## 3. Results

### 3.1. Pharmacological Inhibition of HSP90 with PU-H71 Reduces Glioma Cell Proliferation

Challenges in the treatment of glioma include therapy-resistant stem cells, BBB impermeability, and high intra- and inter-tumor heterogeneity. For this reason, it is crucial to assess the efficacy of drugs against glioma cells with diverse molecular backgrounds with various MGMT promoter methylation statuses [4,5,6,30]. Therefore, we used Glioblastoma IDH wildtype 4 patient-derived glioma stem-like cells and commercial adherent cells with a varied MGMT promoter methylation status to evaluate PU-H71’s activity against glioma cell proliferation. These cell lines included GSC262 (Proneural, unmethylated), GSC811 (Proneural, methylated), GSC11 (Classical, methylated), GSC23 (Classical, unmethylated), GSC20 (Mesenchymal, unmethylated), GSC272 (Mesenchymal, methylated), LN229 (methylated), U251-HF (methylated), and T98G (unmethylated) glioma cells and NHAs (unmethylated). All the glioma cell lines showed sensitivity towards PU-H71. After 72 h of PU-H71 treatment, GSC11, GSC23, GSC272, GSC262, GSC811, LN229, T98G, and U251-HF cells (Figure 1a,b) showed greater sensitivity, with IC50 (half-maximal inhibitory concentration) values of 0.1–0.5 µM, whereas GSC20 cells showed relatively less sensitivity, with an IC50 value of 1.5 µM. NHAs showed the least sensitivity towards PU-H71, with an IC50 value of 3.0 µM (Figure 1c). This value was twice that for the GSC20 cells and 6–30 times that for the other glioma cell lines, indicating that PU-H71 has less of an impact on normal cells than on glioma cells (Figure 1d). In addition, PU-H71’s activity against the cell lines was dose- and time-dependent.

### 3.2. Pharmacological Inhibition of HSP90 with PU-H71 Induces Programmed Cell Death in Glioma Cells

Earlier work from our laboratory showed that HSP90 inhibition with onalespib causes programmed cell death in glioma cells [17,19]. To assess if the pharmacological inhibition of HSP90 by PU-H71 could also induce programmed cell death in glioma cells, we treated GSC811 and U251-HF cells with vehicle control (DMSO) or PU-H71 (0.25 or 1.0 µM), stained them with annexin V–FITC, and subjected them to flow cytometry. Similarly to its impact on cell proliferation, PU-H71 increased the preapoptotic and apoptotic cell population in a dose- and time-dependent manner (Figure 2a). From 48 to 72 h, the percentages of annexin-positive GSC811 and U251-HF cells increased from 8.5% to 13% and from 6.5% to 28.8%, respectively. Increases in the necrotic cell populations reduced the percentages of live GSC811 and U251-HF cells from 85.5% to 73.8% and from 85.2% to 39.9%, respectively.

PU-H71 also induced dose-dependent changes in viable GSC811 and U251-HF cells at 72 h (Figure 2b).

### 3.3. HSP90 Inhibition Negatively Impacts the Survival-Related Biological Characteristics of Glioma Cells

Some of the properties that help glioma cells proliferate and resist treatment are clonogenicity, migration, and invasion. The inhibition of these biological activities by pharmacological inhibitors or miRNA has been shown to sensitize glioma cells to conventional therapies such as temozolomide and radiation [31,32,33]. Therefore, we assessed the colony-forming ability of adherent LN229, T98G, and U251-HF glioma cells treated with vehicle or PU-H71. Compared with the vehicle-treated cells, the PU-H71-treated cells gave rise to 75–90% fewer colonies (Figure 3a).

We used a wound-healing assay to assess the 2-dimensional migration of LN229, T98G, and U251-HF glioma cells treated with vehicle or PU-H71. After 24 h of treatment, the vehicle-treated cells invaded and filled approximately 75% of the scratched space, whereas the PU-H71-treated cells filled only 10–20% of the space, indicating that PU-H71 severely impairs glioma cell migration (Figure 3b). We used a Boyden chamber/transwell migration assay to assess the 3-dimensional migration of HBMECs treated with vehicle or PU-H71. The number of migrated PU-H71-treated cells was approximately 70% lower than that of the migrated vehicle-treated cells, indicating that PU-H71 impairs the migration capability of glioma cells (Figure 3c). Together, these results show that PU-H71 potently inhibits the migration capability of glioma cells.

### 3.4. PU-H71 Induces Programmed Cell Death in Glioma Cells Specifically

The ongoing issue with cancer therapy is its toxic impact on normal cells along with cancer cells [6]. To assess the toxicity of PU-H71, we compared the expression levels of HSP90, HSP70 (a molecular marker of HSP90 inhibition), and cleaved PARP (a molecular marker of programmed cell death) in normal cells with those in glioma cells (Figure 4). In alignment with previously published work [17], HSP90 expression did not differ significantly between normal cells and glioma cells. GSC262 and GSC811 cells and NHAs all showed robust time-dependent increases in their levels of HSP70 expression, indicating the successful pharmacological inhibition of HSP90 by PU-H71. In addition, there was a time-dependent increase in cleaved PARP (Figure 4), indicating the induction of programmed cell death, in PU-H71-treated GSC262 and GSC811 cells but not NHAs, showing that PU-H71 has specificity for glioma cells over normal cells.

### 3.5. PU-H71 Downregulates Pro-Survival Client Proteins in Glioma Cells

Previous work from our laboratory showed that the inhibition of HSP90 with Onalespib causes the downregulation of EGFR and its downstream pro-survival signaling proteins [17,19]. To assess whether PU-H71’s inhibition of HSP90 exerts a similar effect, we treated patient-derived GSC262 and GSC811 cells with vehicle or 0.5–2.0 µM PU-H71 for 7–48 h and used Western blotting to measure the expression of EGFR, P-AKT, AKT, P-MAPK, MAPK, P-S6, S6, cleaved PARP, and GAPDH (loading control). PU-H71 resulted in the robust dose-dependent (Figure 5a) and time-dependent (Figure 5b) downregulation of EGFR, P-AKT, P-MAPK, and P-S6. However, the levels of AKT, MAPK, and S6 were not significantly different between the vehicle- and PU-H71-treated cells, indicating the potent inhibition of HSP90 by PU-H71. In addition, PU-H71 increased cleaved PARP levels in a dose- and time-dependent manner, indicating the induction of programmed cell death due to the downregulation of EGFR and its downstream pathways.

### 3.6. Pharmacological Inhibition of HSP90 with PU-H71 Sensitizes Glioma Cells to Temozolomide

Historically, no single agent alone has yielded significant improvements in outcomes of glioma survival. The current standard of care for glioma is concurrent radiotherapy and temozolomide. The use of combination therapy usually enables a reduction in the therapeutic dose and thus a reduction in toxic effects. HSP90 inhibitors can attenuate DNA repair proteins [5,19,30,34]. Recent studies have shown that DNA repair proteins are involved in temozolomide resistance [7]. Therefore, using patient-derived GSC811 GSCs, we determined the synergistic index of various concentrations of PU-H71 and temozolomide (Figure 6a,b). PU-H71 (0.1–1.0 µM) and temozolomide (200–600 µM) decreased the proliferation rate of GSC811 cells in a dose-dependent manner. We also plotted a dose–response matrix (Figure 6c) and synergistic indices (Figure 6d) using SynergyFinder 3.0 [35]. Higher doses of PU-H71 and temozolomide in combination robustly inhibited cell proliferation. However, 0.5–1.0 µM PU-H71 and 200 µM temozolomide showed a higher degree of synergy, indicating that concurrent PU-H71 could enable a significant reduction in the temozolomide dose. These results indicate that HSP90 inhibitors can sensitize glioma cells to temozolomide.

## 4. Discussion

Ongoing challenges in improving the outcomes of glioma patients include intrinsic and inter-patient tumor heterogeneity, therapy-resistant GSCs, and glioma cells’ high adaptability to a complex and variable tumor microenvironment [4,9,30]. Most preclinical studies have not been translated into successful clinical trials, mainly because of limited BBB drug permeability, the presence of drug efflux pumps, and/or the utilization of inadequate glioma models [6]. To represent intertumoral heterogeneity and drug-resistant glioma cells, we used NHAs, commercial glioma cells (MGMT-methylated and -unmethylated), and patient-derived GSCs with proneural, classic, or mesenchymal molecular classifications and MGMT-methylated or -unmethylated statuses for the evaluation of PU-H71 against glioma. Our study is one of the first to evaluate the efficacy of PU-H71 against patient-derived GSCs. Our results also show for the first time that PU-H71 can overcome inter-tumor heterogeneity and be used against glioma cells with varied molecular backgrounds and differing MGMT promoter methylation statuses. In addition, our results demonstrate the selective activity of PU-H71 against tumor cells compared to normal cells. Our results also show that HSP90 inhibition in combination with PU-H71 potently downregulates multiple pro-survival kinase pathways in glioma, overcoming inter-tumoral heterogeneity.

Challenges in treating glioma include adverse side effects and drug toxicity. Reducing the drug dose and identifying the molecular targets that are differentially regulated between cancer cells and normal cells can help reduce the toxic impact. Chiosis et al. developed PU-H71, a completely synthetic and purine-based HSP90 inhibitor [36,37]. Earlier research demonstrated that PU-H71 has potent antineoplastic effects and a high specificity for cancer cells. The binding affinity of PU-H71 for HSP90 in SKBr3 human breast adenocarcinoma cells was approximately 3 times that in normal heart and lung tissues. PU-H71 also showed a 50-fold higher selectivity for inhibiting malignant cell growth than for inhibiting normal fibroblast growth [8,16,38]. Similarly to previously published work, we observed that glioma cells were more sensitive to PU-H71 than normal cells were. However, the PU-H71 IC50 values for the glioma cells we tested fell within two ranges, 0.1–0.5 µM for sensitive cells, and 1.0–1.5 µm for less sensitive cells. The PU-H71 IC50 value for NHAs was 3.0 µM. The ratio of the IC50 value was twice that for the less sensitive glioma cells and 6–30 times that for the other, more sensitive glioma cell lines. Cell death was significantly lower in NHAs than in glioma cells. Both the ratio of sensitivities and the amount of cell death values are significantly lower than those observed for human breast adenocarcinoma cells, indicating that glioma cells are relatively less sensitive to PU-H71 and would require a higher therapeutic dose of the drug.

We also found that PU-H71 downregulated EGFR and its downstream signaling pathway proteins, including AKT, S6, and MAPK, in glioma cells with and without MGMT methylation. PU-H71-treated GSC262 and GSC811 cells had elevated levels of cleaved PARP, indicating the induction of programmed cell death. This downregulation of the EGFR-AKT-S6 axis is consistent with that observed during the HSP90 pathway inhibition by other HSP90 inhibitors, such as Onalespib [17,19], 17-AAG [22], and 17-DMAG [25].

HSP90 inhibitors have been shown to simultaneously and robustly downregulate several pro-survival kinases in vitro. However, these inhibitors did not successfully translate into clinical applications owing to an inadequate impact on molecular targets, adaptive drug resistance, or drug-related toxicity [8,15]. These issues might be resolved by using combinations of these drugs at lower doses [8,39,40,41]. The results of the present study show that PU-H71 and the DNA-alkylating agent temozolomide (part of the current standard therapy for glioma) work synergistically to reduce glioma cell proliferation. Given this synergistic activity, the effective doses of PU-H71 and temozolomide could both be reduced to reduce toxicity. These results provide a rationale for developing combination therapies with HSP90 inhibitors to treat gliomas.

In the present study, the pharmacological inhibition of HSP90 with PU-H71 potently depleted pro-survival kinases in patient-derived glioma cells; however, in vivo experiments in our laboratory showed that the drug has limited BBB permeability in in vivo glioma models. Our findings regarding PU-H71’s BBB permeability are similar to those of Caldas-Lopes et al. [16], who observed an intratumoral accumulation of PU-H71. They also measured significant concentrations of the drug in the kidneys, liver, lungs, and plasma of mice. However, brain tissue showed minimal PU-H71, indicating that PU-H71 has limited BBB permeability, which limits the use of this drug against glioma [16]. This challenge might be overcome with methods such as magnetic resonance-guided focused ultrasound [42,43], intratumoral drug injection [44], or the delivery of PU-H71 encapsulated in a BBB-permeable nanovesicle [45,46]. Regardless of the limited BBB permeability of PU-H71, our results show that HSP90 inhibition with PU-H71 potently downregulates multiple pro-survival kinase pathways in glioma, overcoming inter-tumoral heterogeneity.

PU-AD, another related epichaperome inhibitor, has demonstrated a selective HSP-90 inhibition in cancer cells and mice, with a similar mechanism as PU-H71. Due to its high brain permeability, this drug was initially tested against Alzheimer’s disease (where Hsp90 is a relevant target) with the completion of Phase 1 (NCT03935568) allowing it to move to Phase 2 clinical trials (NCT04311515) [47,48]. A study of PU-AD in patients with recurrent glioblastoma was initiated to assess the mechanistic similarity and high blood–brain barrier permeability that would be favorable against glioblastoma; however, the study was terminated prematurely due to the sponsoring company’s closure. Additional BBB-penetrant HSP90 inhibitors are in consideration for trials in brain tumor patients.

Our study has several limitations as follows: PU-H71 has poor BBB permeability and is hence not optimal for direct use in glioblastoma patients; due to this issue, our studies were predominantly in vitro and hence, the in vivo efficacy of the agent could not be adequately tested within the scope of this project. Additional approaches such as Ommaya reservoir-mediated intratumoral infusions, BBB disruption techniques, and the use of focused ultrasound may overcome these limitations and could be the focus of future studies. Also, our study did not extensively study other normal cell types such as neurons or other brain resident cells. However, the lack of toxicity in the tested normal cells and the lack of significant toxicity in human studies to date with this agent support the tumor-selective effects of PU-H71.

## 5. Conclusions

Our results strongly suggest that the HSP90 inhibitor, PU-H71, has potent activity in reducing the proliferation, colony-forming ability, and migration of glioma cells but not normal cells. PU-H71 also attenuates EGFR and downstream signaling pathways to induce programmed cell death in both a dose- and time-dependent manner. Moreover, PU-H71 sensitizes glioma cells to temozolomide. Overall, our results provide a strong foundation for the translational development of HSP90 inhibitors for glioma treatment.

## Figures and Tables

**Figure 1 cancers-16-03934-f001:**
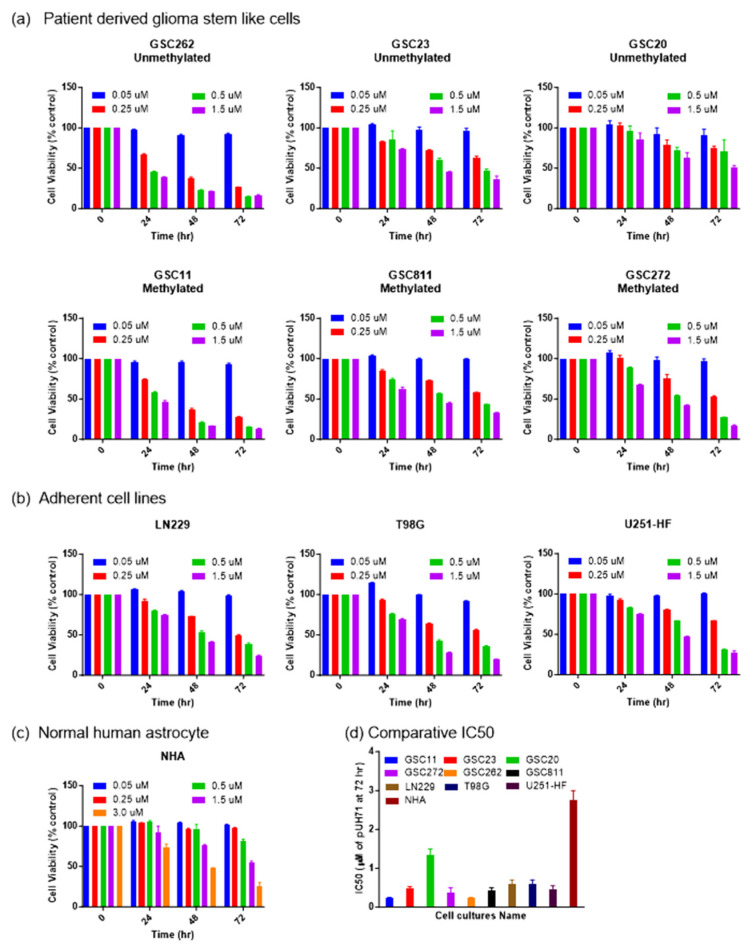
PU-H71 inhibits glioma cell proliferation. (**a**) Percentage cell proliferation rates of GSC11, GSC23, GSC20, GSC262, GSC811, GSC272 (**b**) LN229, T98G, U251-HF (**c**) and NHA after 0, 24, 48, and 72 h of treatment with the indicated concentrations of PU-H71. (**d**) IC50 values for the indicated glioma cell lines and NHAs after 72 h of PU-H71 treatment.

**Figure 2 cancers-16-03934-f002:**
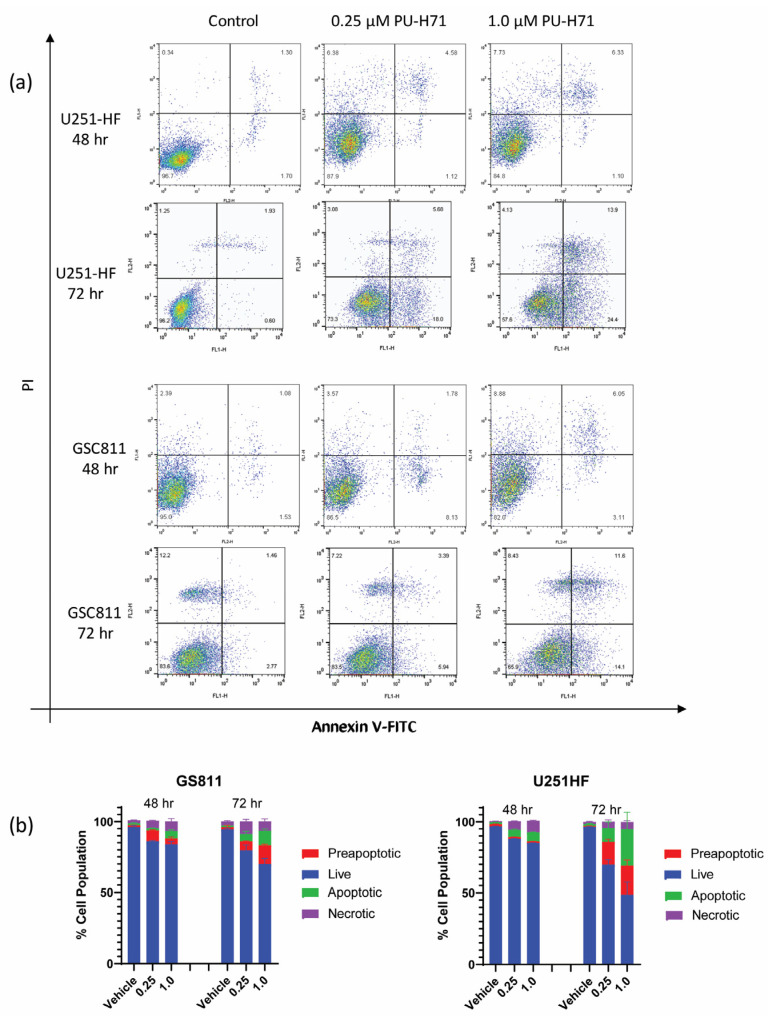
PU-H71 induces programmed cell death in glioma cells. (**a**) Logarithmic displays of flow cytometry data for U251-HF and GSC811 cells treated with vehicle or with 0.25 or 1.0 µM PU-H71 for 48 or 72 h and stained with annexin V–FITC and propidium iodide. (**b**) Proportions of live (blue), preapoptotic (red), apoptotic (green), and necrotic (purple) cells after 48 and 72 h of treatment with vehicle or 0.25 or 1.0 µM PU-H71.

**Figure 3 cancers-16-03934-f003:**
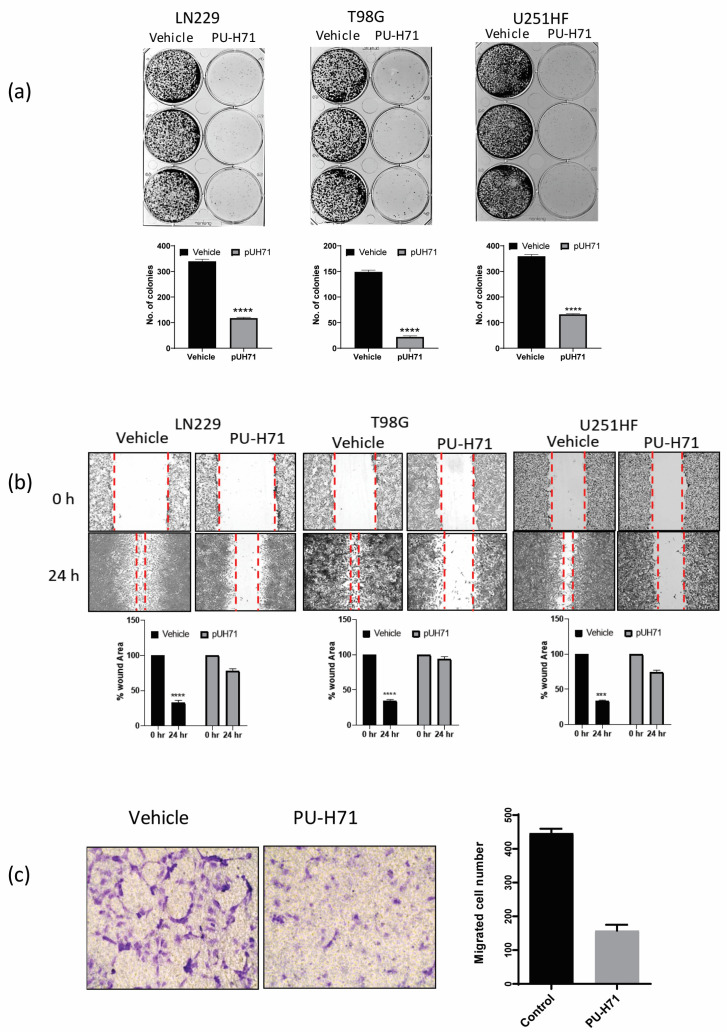
PU-H71 inhibits the biological functions of glioma cells. (**a**) Gelcount^TM^ images show the colony-forming ability of LN229, T98G, and U251-HF cells exposed to vehicle or PU-H71. The graphs show the numbers of colonies. (**b**) Brightfield microscopy images show the wound-healing capability of LN229, T98G, and U251-HF cells exposed to vehicle or PU-H71. The graphs show the percentages of wound areas not enclosed by the cells after 0 and 24 h of PU-H71 treatment. (**c**) Brightfield microscopy images show the migration ability of HBMECs exposed to vehicle or PU-H71 in a transmembrane assay. The graph shows the numbers of migrated cells at 48 and 72 h of treatment with vehicle or 0.25 or 1.0 µM PU-H71. *** *p* < 0.001, **** *p* < 0.0001.

**Figure 4 cancers-16-03934-f004:**
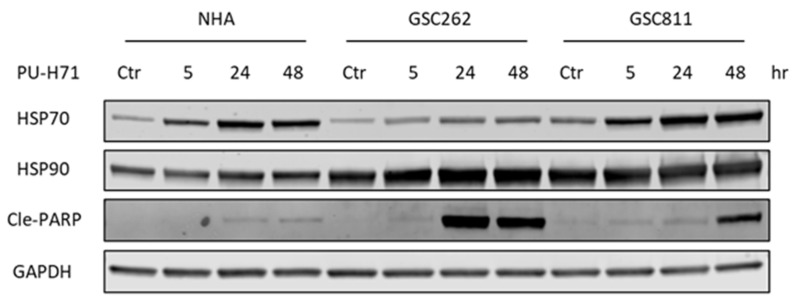
PU-H71 induces programmed cell death in glioma cells specifically. Immunoblotting analysis of the expression of HSP90, HSP70 (a molecular marker of HSP90 inhibition), cleaved PARP (a marker of cell death), and GAPDH (loading control) in vehicle- or PU-H71-treated NHAs and GSC262 and GSC811 cells. Raw images of these data are available in Appendix A.

**Figure 5 cancers-16-03934-f005:**
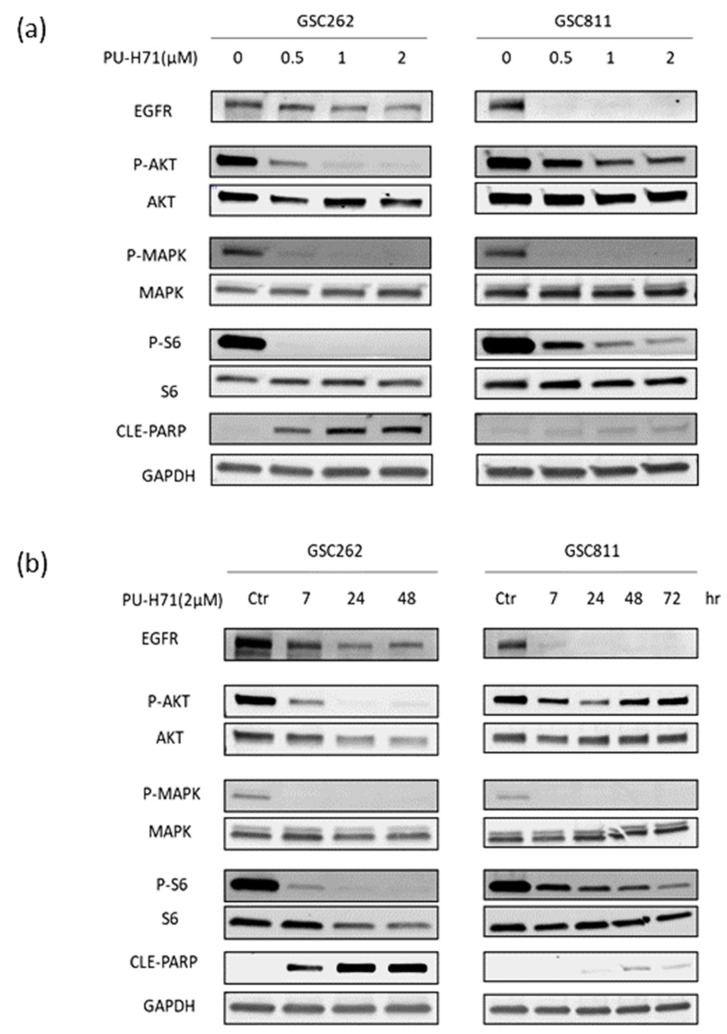
PU-H71 downregulates the expression of HSP90 effector proteins. (**a**,**b**) Immunoblotting analysis showed the dose-dependent (**a**) and time-dependent (**b**) impact of HSP90 inhibition on EGFR, p-AKT, AKT, p-MAPK, MAPK, pS6, S6, and cleaved PARP. GAPDH was used as the loading control. Raw images of Figure 5a and Figure 5b are available in Appendix A, respectively.

**Figure 6 cancers-16-03934-f006:**
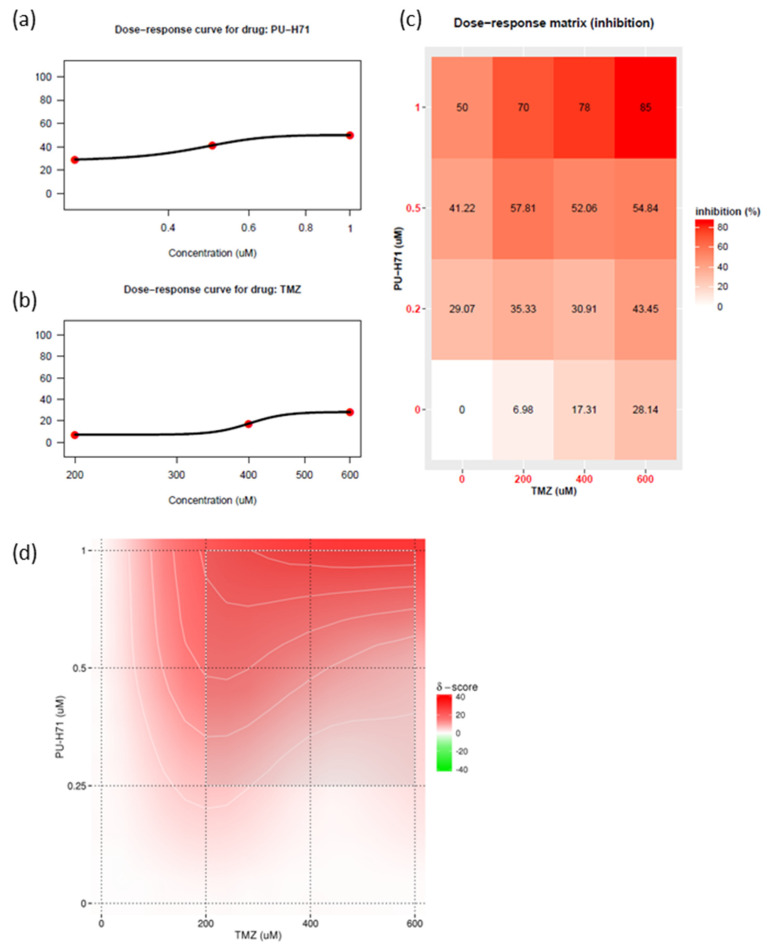
PU-H71 sensitizes glioma cells to temozolomide. (**a**) Dose–response curve for PU-H71. (**b**) Dose–response curve for temozolomide (TMZ). (**c**) Matrix representing the dose-dependent inhibition of cell growth. (**d**) Bliss energy model of the synergistic index for PU-H71 plus temozolomide.

## Data Availability

The data presented in this study are available upon request from the corresponding author.

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
