# Peer review of "Epichaperome Inhibition by PU-H71-Mediated Targeting of HSP90 Sensitizes Glioblastoma Cells to Alkylator-Induced DNA Damage"

_cancers, 2024, doi:10.3390/cancers16233934_

Round 1

Reviewer 1 Report

Comments and Suggestions for Authors

See the attachment.

Author Response

  1. Add a recent literature related to PU-H71 and HSP90. Like, https://doi.org/10.1007%2Fs10637-017-0495-3, https://doi.org/10.18632/aging.205952, https://doi.org/10.3389/fonc.2021.612354

We have now added the reference and results from PU-H71 clinical trial to introduction section.

“Intravenous administration of PU-H71 in patients with refractory solid tumors, over 1 h on days 1 and 8 of 21-day cycles. PU-H71 was well tolerated (10 to 470 mg/m2/day) by patients under this study and no dose limiting toxicity was observed. (https://link.springer.com/article/10.1007/s10637-017-0495-3)”

After careful reading of article “Prognostic relevance and validation of ARPC1A in the progression of low-grade glioma ” https://doi.org/10.18632/aging.205952, we observed that due to difference in direct target, this article will not add significant information to the article and decided to not include in this revised manuscript.

We appreciate reviewer’s suggestion of adding the reference “Inhibition of HSP90 as a Strategy to Radiosensitize Glioblastoma: Targeting the DNA Damage Response and Beyond” https://doi.org/10.3389/fonc.2021.612354. We have edited the introduction section and added this reference.

Combining the conventional therapies with HSP90 inhibitors had shown promising results to sensitize glioma cells towards radiation [https://doi.org/10.3389/fonc.2021.612354], chemo [21]or both [22].

  1. Introduction: Consider adding a section that outlines the translational potential of this work, including any potential biomarkers for patient selection or how these findings align with current therapeutic strategies for GBM.

We have added a section on using an alternate HSP90 inhibitor, PU-AD that has high BBB permeability and inhibit epichaperome. We have briefly mentioned the molecular targets that are known to be impacted by HSP90 inhibitors.

“HSP90 is an ATP-dependent, highly conserved, ubiquitous molecular chaperone that regulates more than 200 client proteins involved in regulating the cell cycle (e.g., CDK4, CDK6), inducing angiogenesis (HIF1A, VEGFR, PI3K/AKT, RTKs), activating downstream pro-survival pathways (EGFR, PI3K/AKT, MAPK), deregulating cellular energies (ARNT, ARRB1, HIF1A, HMG1, SREBF1), and resisting cellular death (NF-kB, AKT, p53, c-MET, APAF1, survivin) [9, 17-19]. The pharmacological inhibition of HSP90 simultaneously downregulates the expression of multiple effector proteins, making HSP90 an attractive therapeutic target in glioma [17, 18, 20-22].”

In the present study we observed the comparable inhibitor effect in patient derived glioma stem like cells, harboring different p53, MGMT promoter methylation status and p16 deletion status etc., indicating there will not be a molecular marker selection criterion for inhibitors in similar category.

  1. There is bulk citation like 3-7, 8-10 and 11-13. Please be specific and add relevant and most recent one only.

We have carefully reviewed the manuscript to remove any non-relevant citations.

  1. Include a paragraph on: How this research work is different then previous published work?

Keeping the flow of the paper and journal format, we have discussed the novelty of our work in several discussion paragraphs. Some of the discussed points are –

  1. This is the first study to test the efficacy of PU-H71 in patient-derived GSCs.
  2. Our results show that PU-H71 can overcome inter-tumoral heterogeneity and be used against glioma cells with varied molecular classifications and MGMT promoter methylation status.
  3. Our results showed the selective activity of PU-H71 against cancer cells in comparison to normal cells.
  4. Our results also showed that HSP90 inhibition with PU-H71 potently downregulates multiple pro-survival kinase pathways in glioma, overcoming the inter-tumoral heterogeneity.

  1. Future prediction or additional study: While the study demonstrates promising results in preclinical models, it would benefit from a more detailed discussion of how these findings might translate to clinical settings.

Due to limited BBB permeability of PU-H71, it may not be a suitable candidate for clinical trials against gliomas. However, there is an epichaperome inhibitor with high BBB that could be explored for therapeutic potential of HSP90 and epichaperome inhibitors.  We have added this section at the end of discussion.

“PU-AD, also an epichaperome inhibitor has demonstrated selective HSP-90 inhibition in cancer cells and mice with similar mechanism as of PU-H71. Due to its high brain permeability, this drug was initially tested against Alzheimer’s disease (where Hsp90 is a relevant target) and a phase 1 completed (NCT03935568) allowing it to move to Phase 2 clinical trials (NCT04311515) [52, 53]. Combining the mechanistic similarity with high blood brain barrier permeability, a study of PU-AD in patients with recurrent glioblastoma was initiated but was subsequently stopped due to the sponsoring company’s closure.”

  1. Figure 4 and 5., clarity need to improve

Thank you for pointing this out. We have now uploaded 300 dpi images for publication purpose.  

  1. Line 366-368: This challenge might be overcome with methods such as magnetic resonance guided focused ultrasound, intratumoral drug injection, or the delivery of PU-H71 encapsulated in a BBB-permeable nanovesicle., Please add cite or justification for this sentence.

We have now added references on possible ways to use the advance technology such as magnetic resonance guided focused ultrasound, intratumoral drug injection and nanovesicles to deliver drugs across BBB.

  1. Line 373: Our results strongly, is strongly appropriate over here?

Thank you for this comment. Using several experiments to investigate the effect of PU-H71 on glioma biological characteristics and molecular changes, we have unequivocally shown potent effects of PU-H71 against a large panel of glioma cells. This includes cell proliferation assays, flow cytometry-based assay (annexin-PI and cell cycle change by PI) and western blot (staining with cleaved-PARP). Given the robust activity in patient derived glioma cell lines in several types of experiments using multiple cell lines that confirms that these are indeed drug induced and glioma targeting effects through multiple lines of evidence, we hope the reviewer will agree that our statement is not a hyperbole but a evidence-supported statement of certainty of these effects and may be appropriate under these circumstances

Reviewer 2 Report

Comments and Suggestions for Authors

The study described that pro-survival signals increased glioma cells’ sensitivity to temozolomide, and the combination of PU-H71 and temozolomide had greater anticancer efficacy than either agent alone. In addition, I will suggest you to highlight the novelty of your work. Kindly add recent literature to discuss your job efficiently. I suggest you improve your future perspectives in more detail by adding possible methodology. in addition, how authors look at nanotechnological aspect of this work in future please discuss. Kindly add one pictorial repersentation of your proposed mechanistic insight of this work. 

Author Response

The study described that pro-survival signals increased glioma cells’ sensitivity to temozolomide, and the combination of PU-H71 and temozolomide had greater anticancer efficacy than either agent alone. In addition, I will suggest you highlight the novelty of your work. Kindly add recent literature to discuss your job efficiently. I suggest you improve your future perspectives in more detail by adding possible methodology. in addition, how authors look at nanotechnological aspect of this work in future please discuss. Kindly add one pictorial representation of your proposed mechanistic insight of this work. 

Thank you for your valuable input in reviewing this article.

We have now highlighted the novelty of the work and its differences from previous reports HSP90 inhibitors, their targets, and manuscripts related to glioma in the discussion section as follows-

  1. This is the first study to test the efficacy of PU-H71 in patient-derived GSCs.
  2. Our results show that PU-H71 can overcome inter-tumoral heterogeneity and be used against glioma cells with varied molecular classifications and MGMT promoter methylation status.
  3. Our results showed the selective activity of PU-H71 against cancer cells in comparison to normal cells.
  4. Our results also showed that HSP90 inhibition with PU-H71 potently downregulates multiple pro-survival kinase pathways in glioma, overcoming the inter-tumoral heterogeneity

We have reviewed our manuscript again to ensure that relevant research is cited.

For future directions-

Due to limited BBB permeability of PU-H71, it may not be a suitable candidate for clinical trials against gliomas. However, there is an epichaperome inhibitor with high BBB that could be explored for therapeutic potential of HSP90 and epichaperome inhibitors.  We have added this section at the end of discussion.

“PU-AD, also an epichaperome inhibitor has demonstrated selective HSP-90 inhibition in cancer cells and mice with similar mechanism as of PU-H71. Due to its high brain permeability, this drug was initially tested against Alzheimer’s disease (where Hsp90 is a relevant target) and a phase 1 completed (NCT03935568) allowing it to move to Phase 2 clinical trials (NCT04311515) [52, 53]. Combining the mechanistic similarity with high blood brain barrier permeability, a study of PU-AD in patients with recurrent glioblastoma was initiated but was subsequently stopped due to the sponsoring company’s closure.”

We have added references on possible ways to use the advance technology such as nanotechnology to deliver drugs across BBB.

We have now added a figure as graphical abstract summarizing the PU-H71 inhibitor mechanism.

Reviewer 3 Report

Comments and Suggestions for Authors

Dear authors,

congratulations on your work, however some changes need to be made before it can be considered for publication.

1) The introduction should contain more information about GBM, and its classification, read and cite:  https://doi.org/10.3390/biomedicines12010008

2) The description of the experimental aspect could benefit from a graphic illustration of the process

3) the discussion should be fractionated and reinforced in terms of bibliography

4) read and cite  https://doi.org/10.3390/brainsci14030296

5) you need to write the limitations before the conclusion

Author Response

  1. The introduction should contain more information about GBM, and its classification, read and cite:  https://doi.org/10.3390/biomedicines12010008

Thank you for your suggestion. The reference cited by the reviewer is related to neurosurgical techniques in gliomas and not to GBM classification. However, we agree that it would be useful to cite relevant references related to glioma classification. Hence, we have instead cited the original publication related to WHO 2021 classification of gliomas . (doi: 10.1093/neuonc/noab106).

2. The description of the experimental aspect could benefit from a graphic illustration

Thank you for your valuable suggestion. We have now added a figure as graphical abstract summarizing the PU-H71 inhibitor mechanism.

3. the discussion should be fractionated and reinforced in terms of bibliography

We have carefully reviewed the manuscript to remove any irrelevant citations.

4. read and cite  https://doi.org/10.3390/brainsci14030296

We appreciate reviewer for pointing out this article “Clustering Functional Magnetic Resonance Imaging Time Series in Glioblastoma Characterization: A Review of the Evolution, Applications, and Potentials”. However, we are uncertain how this imaging data analysis paper focusing on functional MRI and other such techniques is relevant to our manuscript on HSP90 inhibitors. Perhaps the reviewer meant to provide a different reference related to our topic? If so, pls let us know and we will be happy to consider the same.

5. you need to write the limitations before the conclusions.

We agree with reviewer’s suggestion of writing limitations before the conclusions. Due to journal’s format, we do not have a separate section for the limitation. We have discussed some of the limitation of PU-H71 at the end of discussion section. 

“Our findings regarding PU-H71’s BBB permeability are similar to those of Caldas-Lopes et al. [19], who observed an intratumoral accumulation of PU-H71. They also measured significant concentrations of the drug in the kidneys, liver, lungs, and plasma of mice. However, brain tissue showed minimal PU-H71, indicating that PU-H71 has limited BBB permeability, which limits the use of this drug against glioma [19].

However, we have added additional limitations of this study in the discussion section now.

“Our study has several limitations: PU-H71 has poor BBB penetrance and is hence not optimal for direct use in glioblastoma patients; due to this issue, our studies were predominantly in vitro and hence, the in vivo efficacy of the agent could be adequately tested within the scope of this project. Additional approaches such as Ommaya reservoir- mediated intratumoral infusions, BBB disruption techniques and use of focused ultrasound may overcome these limitations and can be the focus of future studies. Our studies also did not widely study other normal cell types such as neurons or other brain resident cells. However, the lack of toxicity in the tested normal cells and the lack of significant toxicity in human studies to date with this agent support the tumor-selective effects of PU-H71.”

Round 2

Reviewer 1 Report

Comments and Suggestions for Authors

Thank you. All suggession incorporated. 

Author Response

Thank you. All suggession incorporated. 

We thank the reviewer for your acceptance of our responses with no additional requirements and appreciate your time and suggestion to improve our manuscript.

Reviewer 3 Report

Comments and Suggestions for Authors

The response to my directions was only partial, you did not read the first paper properly, there is a large section on classification that takes into account some changes even after the 2021 classification. Some of your directions in the paper are wrong, please read that manuscript carefully. Also in that manuscript there are important indications about the molecular part as well as the surgical aspect, not focused only on the site but read the topic well.

The other article on fMRI, It is as good a suggestion as any to give the manuscript a more clinical focus.

Please handle the requested revisions completely, not interpret them, otherwise there would be no point in this process.

Author Response

Response to the directions was partial. There is a large section on classification that takes into account some changes even after the 2021 classification. That manuscript also has important indications about the molecular part as well as the surgical aspect. The other article on fMRI, It is as good a suggestion as any to give the manuscript a more clinical focus. Please handle the requested revisions completely, not interpret them.

After a thorough reading of the suggested review articles again which are relevant to surgical and imaging aspects of the field (and which we note are from the same set of authors), per the reviewer’s suggestion, we feel that they are of little relevance to this work and do not add to our pre-clinical studies on targeted inhibition of HSP90 in epichaperome complex by a novel HSP90 inhibitor, PUH-71, which is the focus of our paper. We appreciate the reviewer’s comments and suggestions and in our prior revision had addressed the same by referring to primary articles that highlight the background of glioma classification and biology. Hence, we would like to retain the current references and make no additional changes.